# Recent Developments of Femtosecond Laser Direct Writing for Meta-Optics

**DOI:** 10.3390/nano13101623

**Published:** 2023-05-12

**Authors:** Shuai Xu, Yangfan Zhang, Ting Wang, Le Zhang

**Affiliations:** 1College of Intelligent System Science and Engineering, Shenyang University, Shenyang 110044, China; 2Shenyang Institute of Automation, Chinese Academy of Sciences, Shenyang 110016, China

**Keywords:** femtosecond laser, meta-optics, polarization convertor, geometric phase optics, nanogratings

## Abstract

Micro-optics based on the artificial adjustment of physical dimensions, such as the phase, polarization, and wavelength of light, constitute the basis of contemporary information optoelectronic technology. As the main means of optical integration, it has become one of the important ways to break through the future bottleneck of microelectronic technology. Geometric phase optical components can precisely control the polarization, phase, amplitude and other properties of the light field at the sub-wavelength scale by periodically arranging nanometer-sized unit structures. It has received extensive attention in the fields of holographic imaging and polarization optics. This paper reviews the physical mechanism of micro-nano structure modification, research progress of femtosecond laser direct-writing photoresist, femtosecond laser ablation of metal thin films, femtosecond laser-induced nanograting, and other methods for preparing polarization converters and geometric phase optics. The challenges of fabricating ultrafast optical devices using femtosecond laser technology are discussed.

## 1. Introduction

Micro-nano structures have unique optical, electrical, magnetic and thermal properties. Research on the preparation of micro and nano structures has considerable value and broad development prospects. The preparation of micro and nano structured materials is inseparable from the advanced micro and nano processing technology [1,2]. A femtosecond laser is an ultrashort laser pulse with pulse duration usually in the order of tens to hundreds of femtoseconds. It is capable of emitting high-intensity pulses in a very short period of time, which can precisely ablate materials and form small heat-affected zones; thus, it has a high nanometer resolution and precision. The advantages of using deterministic and reproducible femtosecond laser processing allow it to process almost any material, but currently the preparation of geometric phase components in some difficult materials can be a great challenge, such as diamond [3,4,5]. Moreover, the laser pulse parameters, such as energy and duration, are significant when inducing different structures with functions in fabrication. For pulse duration electron–phonon coupling time, the energy transfer time is much shorter than the heat diffusion time, and with a relatively low pulser energy, leads to a smooth and modified carrot-shaped region without apparent damage characterized by a uniform positive change, and higher energy brings about self-ordered nanogratings with birefringence. In contrast with the former non-damage situation, for pulse duration electron–phonon coupling time, the electrons, and lattice reach thermodynamic equilibrium on the timescale of the pulse duration, thus causing damage to the fabrication area via energy accumulation. Overall, femtosecond lasers show great application advantages for microstructure and geometric phase device processing [6,7].

Since the birth of geometric optics, researchers have designed the contours of natural materials to change the optical path difference and realize the regulation of light transmission behavior, such as lenses. Geometric optical devices have simple design and stable functions, but it is difficult to realize certain functions, such as phase distribution regulation through simple design, and it is difficult to achieve miniaturization due to the lack of certain precision. Wave optical devices based on diffraction principle, such as flat-top beam generator and beam homogenizer, are widely used in laser processing, biomedicine, communication and sensing due to their high-energy efficiency and accuracy [8,9,10]. With the advancement of micro-nano processing technology, researchers are also exploring sub-wavelength devices, such as meta-surfaces and photonic crystals, which have more potential for miniaturization and integration [11,12,13,14]. Recently, geometric phase devices as well as polarization converters have become a new choice for the spatial modulation of phase and polarization [15,16,17,18,19]. Without changing the device thickness and refractive index, such devices achieve continuous modulation by adjusting the orientation of the nanocells, thus creating integrated devices with high diffraction efficiency [20,21,22]. Meanwhile, femtosecond laser direct writing has been widely applied to refined fabrication due to its non-contact processing and efficient material removal [23,24,25]. High-resolution fabrication of tens of nanometers on transparent materials and semiconductor thin films have been achieved thanks to nonlinear absorption and localized light enhancement [26,27]. In this paper, the recent progress of femtosecond laser direct-writing polarization converters and geometric phase devices on different materials is reviewed. As a key element in the fabrication process, the forming mechanism of femtosecond laser-induced nanogratings is discussed. Furthermore, some challenges of fabricating optical devices by a femtosecond laser are included at the end.

## 2. Fabrication Principle of Polarization Converter and Geometric Phase Devices

In 1892, Poincare proposed a useful method to describe polarization, that is, in spherical coordinates, the basic polarization state is mapped by Stokes parameters to the surface of a sphere, which is called the Poincare sphere. Figure 1A shows that the Poincare sphere can represent a uniform polarization state intuitively. Every point on the Poincare sphere has a one-to-one correspondence with all possible polarization states, and the modulation of polarization can be regarded as the point movement on the Poincare sphere. In 1956, Pancharatnam et al. found that an electromagnetic wave of a certain polarization state changes along a closed path on the surface of the Poincare sphere, and there exists a phase difference between the final state and initial state which is equal to half of the solid angle of the closed path. Since the phase factor is only related to the geometric path of polarization evolution, it is called the geometric phase, also known as the Pancharatnam–Berry phase (PB), for Berry also contributed to the summary of the theory.

The geometric phase can be easily observed in such condition: a circular polarized light goes through a geometric phase device, which mostly consists of spatial-distributed anisotropy/birefringence elements, and the phase retardance of the orthogonal-polarized components is determined by the orientation angle of anisotropy elements. It can be proved by the Jones matrix calculation that when a circular polarized light passes through a birefringence structure (crystal, nano-metal rod, nano-grating, etc.) in a specific optical axis direction, an orthogonal polarization state will be generated with twice the phase change as the optical axis angle. For example, the grating structure shown in Figure 1B has different azimuth angles θ. This anisotropic structure exhibits the function of birefringence (see Section 2 for detailed derivation) and thus has different Jones matrices $J_{\mathrm{wp}}$. The orthogonal polarization state generated after circular polarization incident contains the $e^{i2\theta }$ term, and the azimuth-dependent phase change.
(1)$$J_{\mathrm{wp}}\frac{1}{\sqrt{2}}\left[\begin{array}{c} 1\\ i \end{array}\right]=\frac{1}{\sqrt{2}}\left[\begin{array}{cc} e^{-i\Gamma /2}\cos^{2}\theta +e^{i\Gamma /2}\sin^{2}\theta & -\mathrm{isin}\left(\frac{\Gamma }{2}\right)\sin2\theta \\ -\mathrm{isin}\left(\frac{\Gamma }{2}\right)\sin2\theta & e^{-i\Gamma /2}\sin^{2}\theta +e^{i\Gamma /2}\cos^{2}\theta \end{array}\right]\left[\begin{array}{c} 1\\ i \end{array}\right]=\frac{1}{\sqrt{2}}\left[\cos\left(\frac{\Gamma }{2}\right)\left[\begin{array}{c} 1\\ i \end{array}\right]-\mathrm{isin}\left(\frac{\Gamma }{2}\right)e^{i2\theta }\left[\begin{array}{c} 1\\ -i \end{array}\right]\right]$$

Therefore, spatially varying polarization state and phase modulation can be realized by the design of spatially varying birefringence structure (including the optical axis angle and the length in the propagation direction). Structures such as gratings and liquid crystals have birefringent properties and are often used as basic elements to form polarization converter and geometric phase devices. By designing the orientation and other parameters of the spatial distribution, the integrated devices are fabricated on different materials with excellent performance.

The geometrical phase plane lens fabricated by Harvard University has the capability of chiral imaging and spatially resolved chiral spectrum. The diagram for chiral imaging is shown in Figure 2A [16]. The rotation of the nanostripes according to the Pancharatnam-Berry phase provides the required phase for focused light. The lens is made up of two nanostripes in blue and green. The focusing efficiency is highest when each nanostripe is used as a half-wave plate to convert circularly polarized input light into transmitted light with opposite helicity. Yang et al. designed an optical vortex plate, in which the stripes are distributed to different discrete sectors by azimuth [18]. A set of silicon nanostripes were designed to keep a retardance δ ≈ π in a very broad spectral range, and with near unit reflection efficiency in Figure 2B. After rotating π/2 of each nanostripe, an additional phase was achieved.

With the development of science and technology, the material structure enters the micro and nano scale when endowed with new properties different from those of individual materials. Advanced micro-nano processing technology is involved in the preparation of micro-nano structured materials, especially femtosecond laser processing, which has been proven to be a micro and nano processing technology with superior performance. Femtosecond laser micro-nano machining has the following advantages: (1) simpler equipment—no need to have clean room facilities; (2) ability to process almost all types of materials; (3) this process is applicable to the surface of any three-dimensional object; and (4) different micro-nano structures can be easily obtained by adjusting the respective parameters. Nowadays, femtosecond laser manufacturing is widely used in almost all types of materials and has attracted extensive attention due to its unique three-dimensional (3D) processing capability, high manufacturing resolution, and scalability.

## 3. Fabrication of Spin-Orbit Angular Momentum Couplers by Femtosecond Laser Two-Photon Direct Writing Technique

Light can carry both energy and momentum, and the momentum of light includes both linear and angular components. Near the paraxial axis, the angular momentum of light has two parts: one is the spin angular momentum related to the circular polarization, and the other is the orbital angular momentum related to the wave vector or phase wavefront direction. There is an interaction between the spin of light and the orbital angular momentum, and the change in the orbit is always accompanied by a lateral spin-dependent deflection. The orbital angular momentum of light endows more dimensional manipulation of light, and researchers have witnessed the remarkable development of vortex light in many fields. A beam of light or a photon can carry two different angular momentums. The orbital angular momentum arises from spin-orbit interactions induced by polarization manipulation. The spin-orbit angular momentum coupler is designed in terms of spin angular momentum (SAM) and orbital angular momentum (OAM) exchange, which occurs in heterogeneous anisotropic media, and its most direct application is the ordinary Gaussian beam starts to generate an OAM-bearing beam [24]. According to Figure 3A, theoretically, when passing through a q-plate with 100% polarization converter efficiency, and with a retardance of δ ≈ π, the polarization state will be completely converted to the opposite state.

As shown in Figure 3B(a), the grating can be seen as two different materials (a, b) stack, and the interface of these two kinds of material meet the boundary conditions of $E_{a∥}=E_{b∥}$ and $D_{a⊥}=D_{b⊥}$, so we can use $\varepsilon_{∥}=\frac{D_{∥}}{E_{∥}}={f\varepsilon }_{a}+\left(1-f\right)\varepsilon_{b}$ and $\varepsilon_{⊥}=\frac{D_{⊥}}{E_{⊥}}=\frac{\varepsilon_{a}\varepsilon_{b}}{{f\varepsilon }_{b}+\left(1-f\right)\varepsilon_{a}}$ to get parallel and perpendicular to the direction of the optical axis of a different nature. Clearly, axial refractive index (no, ne) is different, namely the birefringence $\Delta n=n_{∥}-n_{⊥}$. Its size is related to the grating duty ratio W=f/λ (maximum when W = 0.5). At the same time, we know that the retardance caused by birefringence is related to the thickness (height) of the material in k direction, and that the optimal height h* satisfies the optimal birefringence phase; therefore, Δ = π can be obtained by h* = λ/(π|dn |) to calculate, see Figure 3B(b) for the calculation results.

The femtosecond laser, undoubtedly, has become a common fabrication method for nano structures in photoresist. Wang and his team used femtosecond laser direct writing to fabricate geometric phase optical elements in photoresist for structured light generation [28]. A series of spin-orbit angular momentum couplers were also constructed and characterized. Figure 3C shows the sample characterization under a scanning electron microscope (SEM) after development, rinsing, and drying. δ-illuminates the sample with polarized light, and then performs polarization imaging of the intensity distribution excited by the sample. As can be seen, light with a chirality dependent on helicity is generated in Figure 3D. The spin-orbit interaction was enabled by a lower refractive index material with photoresist. A retardance of π is possible to be achieved based on the subwavelength gratings that are patterned on photoresist. However, geometric phase optics are not restricted to spin-to-orbit angular momentum couplers. They also applied femtosecond laser direct-writing technology to a discrete optical spin beam splitter that can deflect different helicity light. Its scanning electron microscope (SEM) image is shown in Figure 3D, and such a device has a distribution of one-dimensional grating orientations. They are able to generate optical vortices with topological charges |ℓ| = 1, 10, 20, respectively. Colors on elements represent geometric phase modulations caused by changes in its anisotropy variation. Geometric phase optics with their unique advantages (subwavelength thickness, planar shape factor, compatibility with micromachining and nanofabrication techniques, and low cost for bulk manufacturing) have been able to gradually replace conventional kinetic element systems.

## 4. Femtosecond Laser Ablation of Gold Films

An optical vortex generator (q-plate, helical phase plate) produces a beam with a helical wavefront, which can be used for the structural processing of materials and optical polymerization [28,29,30]. The device is realized by using optical anisotropic materials such as liquid crystal, semiconductor, glass, and metal, or dielectric surfaces of different designs [31,32,33,34]. It was previously difficult to fabricate a q-plate with a wavelength of 1.5 m and a shorter visible wavelength by laser printing, since it was difficult to fabricate grating patterns with a high aspect ratio of 0.5 with a high duty ratio and high precision [19]. Although nanotrenches fabricated on opaque thick metal films (50–100 nm) have low transmission efficiency, the resulting optical vortices have substantial purity compared to those produced by well-established nanolithography and plasma etching. It is more practical to explore the possibility of rapid laser fabrication of binary q-plates by ablating metal trenches to achieve higher efficiency and purity, as well as more complex structures [35].

As shown in Figure 4A, Hua et al. realized the rapid fabrication of q-plate by femtosecond laser ablation of the gold film on the glass substrate (using galvanometric scanners with a writing speed of 5 mm/s) [36]. The q-plate made by 50 nm thick gold film ablation is 3% spin-orbit conversion efficient, and the device can be annealed to 800 °C. The laser ablation pattern of q = 3/2 wave plate composed of n = 16 grating segments is shown in Figure 4B. The diameter of the wave plate is 0.4 mm, the period is 0.8 μm, and the notch w is 250 nm. The adjacent structures with 1 micron achieve high fidelity. When the direction of the grating is linearly polarized with respect to the illumination light pi/4, due to formal birefringence, the strongest intensity can be observed and the number of bright lobes is 4q, as shown in Figure 4C. The uniformity of the side lobes shown in the SEM diagram proves that each part of the q-plate has the same ablative quality during the preparation process.

Xu et al. further studied the influence of the thermal annealing process on structure preparation by combining the advantages of efficient processing and the thermal smoothing effect [37]. Blazed diffraction gratings are one of the most dominant geometric phase (GP) devices with the ability to steer a beam of light to several diffraction orders. The grating consists of an array of nano-slits with a simple phase curve that is linearly related to its position, varying by 2π within a period, and such a device can achieve the function shown in Figure 4D for incident light with different polarizations. The design of subwavelength periodic orientation introduces retardance variation. They also fabricated a multi-level GP diffractive focusing lens, which has a multi-level discrete spherical phase function with discrete series N = 4, 8 (Figure 4D). After thermal annealing, the structure surface was smooth and large gold particles and debris were removed, and the intensity distribution of the lens at the focal plane is obviously changed, and the intensity after annealing is much stronger than before annealing. The cross-section of the intensity map shows that the intensity enhancement after thermal annealing is close to 100%, which is due to the higher surface quality of the nanoparticles and the reduction of gold particles. The use of the thermal annealing process completely solves the problem of large roughness and insufficient geometric integrity in surface processing, and the surface shape of the geometric phase is significantly improved after fine-tuning by the thermal annealing process.

Using this technique, as shown in Figure 4F, we fabricated gratings, super-surface PB lenses, q-plates, and “M” holograms, and verified the design performance by analyzing their phases at 808 nm wavelength. The demonstration and validation of the above GP element devices fully demonstrate the feasibility of our novel preparation technology process approach. The efficiency and performance of this method pave the way for the preparation of GP elements with low loss, high-temperature resistance, novel polarization functions and a high-phase gradient, which have the potential for a wide range of applications.

## 5. Femtosecond Laser Direct Writing of Glass-Induced Nanogratings

Since Birnbaum first observed nanostripes in 1965, nanostripes have attracted people’s attention, no matter from the perspective of a physical mechanism or practical application. By adjusting the wavelength and irradiation conditions of the laser, people can partially control the morphology of nanostripes produced by self-organization. The material is irradiated with linearly polarized nanosecond or femtosecond laser pulses to form periodic surface structures. It is understood that such structures are formed by the interference of incident laser light with reflected light. Therefore, fringes are usually oriented perpendicular to the incident polarization.

Recently, many researchers have focused on the processing of solid transparent materials by femtosecond laser. Owing to its higher durability and damage threshold, the geometric phase optical element that has been fabricated in solid materials is of great potential in many prospects, in which, an ultrafast laser also plays an important role. The interaction between transparent materials (such as silica glass) and an ultrafast laser leads to different types of modification, including self-assembly nano-gratings with unique birefringence. Beresna et al. demonstrated the fabrication of spatially varying polarization converters by femtosecond laser-induced, self-assembled nanogratings [38]. The manipulation of birefringence is achieved by controlling the direct writing parameters, especially the polarization azimuth of the direct writing beam. Furthermore, femtosecond laser-induced nanogratings have been demonstrated in a variety of glasses besides fused quartz [39,40,41,42,43,44,45]. Zhang et al. reported in the article that the pulse energy to reach the maximum delay in GeO_2_ glass is ~65% lower than that in fused silica [39]. Under the same processing conditions, the laser-induced delay is twice that of fixed irradiation. The optimal pulse duration for the maximum delay in GeO_2_ glass is in the sub picosecond region, usually around 500 fs, while the fused silica is at the picosecond level, about 1–2 ps. Figure 5A shows the device characterization results, where the retardance image exhibits a relatively uniform distribution with an average value of approximately 0.5 π at a wavelength of 532 nm, which corresponds to a quarter-wave plate value. The optical loss caused by light scattering of nanograting structures is a major drawback, limiting their wider application. Alkali-free alumino-borosilicate glass has a better chemical stability, electrical insulation, mechanical strength and lower coefficient of thermal expansion, and is a special and toughened industrial glass in large quantities. S. S. Fedotov et al. used a femtosecond laser to induce self-assembled nanostructures on alkali-free alumino-borosilicate glass AF32 glass (Figure 5B) [45]. In the experiments, AF32 glass was compared with two other glasses (Boroflfloat 33 and silica) and it was found that the retardance produced by AF32 glass was higher than the other two under the same processing conditions, and the retardance produced by the induced nano-grating structures was much larger than those of silica glass by three orders of magnitude. From the analysis of the birefringence exhibited by the AF32 nanostructures, it was observed that the magnitude of the retardance was related to the glass transfer temperature rather than to the SiO_2_ content in the glass. Finally, polarized vortex beam microconverters were prepared on AF32 glasses that can be used in the near-infrared spectral range. 

Sakakura et al. report a novel ultrafast laser-induced structure (Type X) in silica glass, which consists of randomly distributed nanopores extending perpendicular to the polarization direction of laser, which is also a controlled birefringence structure; however, with the transmittance of up to 99% in the visible and near-infrared, and over 90% in the UV range down to 330 nm [23]. To find out where the low-loss birefringence modification came from and how it changed with the number of pulses (Np), the researchers used methods like SEM, birefringence, and transmission images. When Np = 50, the researchers found that the birefringence modification changed with the number of pulses. With the increase in pulse number (Np = 120–150), the density of the nanopore increases and is elongated perpendicular to the polarization direction. Meanwhile, the retardance increases with the rising nanopore density (Np = 150 is about 5 × 10^−4^). At Np = 200 or so, the nanopores form the nanoplane.

An interesting phenomenon is that the diameter of the modified region in the retardance images (1 μm) is much smaller than the diameter of the beam spot at the focal point (~5 μm). This shows that the modification only happens in the central part of the focused beam because of multiphoton absorption, which is a characteristic of Type X. Figure 5C shows the applications of the above findings. Sakakura et al. fabricated a geometric phase prism with a constant gradient of birefringence slow axis distribution along the horizontal direction. The propagation direction of light can be changed by changing the circular polarity. This lens was also fabricated with a parabolic slow axis distribution, and the focus and blur can also be changed by changing the direction of the circular polarization. Another important application is polarization converters. The retardance of this device is chosen to be one-half of a wavelength, and the slow axis varies linearly from 0° to 180°. The converter is seemingly highly transparent, but it can be clearly observed in the crossed polarizer. The device produces a high-quality 343 nm donut-shaped beam with radial and azimuthal polarization, and the converter has a transmittance of up to 91%. To conclude, the Type X devices have much potential in ultra-low-loss birefringent optical devices.

## 6. Nanograting Induction Mechanism

While using femtosecond laser-induced nanogratings for applications, researchers are also actively exploring their forming mechanisms. Physical models such as plasmonic interference, asymmetric evolution of nano-plasmons, and exciton-pole self-trapping interference are considered to be the reasons for nanograting formation [46,47,48,49]. In 2003, when Shimotsum et al. observed the nanograting structure with SEM for the first time (in Figure 6A), they proposed its formation mechanism [32]. They believe that in the focused region of the femtosecond laser, a large number of electron plasmas would appear due to the nonlinear ionization effect, and under the coupling effect of the light field at the tail end of the incident pulse, the electron plasmas then underwent resonant absorption, leading to the phenomenon of mutual interference between the incident light field and the electron plasmas.

The transmission ranges of specialty glasses, such as chalcogenide glasses, have been synthesized with broad transmission ranges. It has unique physical and chemical properties and a transparent window well beyond the 2–5 μm spectral range of conventional glass. It offers particular advantages for various photonic applications, including space optical communication. Yves Bellouard et al. used a femtosecond laser to induce self-organized nanostructures on and near the surface of Ge_23_Sb_7_S_70_ glass [46]. Figure 6B shows crossed polarizer images of femtosecond laser-induced modification. Formal birefringence effects are due to the presence of periodic structures with alternating refractive indices, showing polarization anisotropy in the figure. The surface of the sample was polished and then etched with NaOH. Figure 6B shows the SEM images under different laser polarizations. The subwavelength periodicity decreases with the increase in pulse energy. At lower energy densities, the laser-induced regions show both positive and negative refractive index changes. At higher fluence levels, the formation of self-organized nanograting structures at and below the surface with structures perpendicular to the laser polarization can be observed, similar to those found in other glasses.

Li et al. proposed a nanowire generation and stretching model by 2020, which could explain the formation mechanism of nanogratings to some extent [26]. First, femtosecond laser-induced multiphoton absorption creates nanovoids, as predicted by near-field enhancement. The nanovoids eventually evolve into dumbbell-shaped or uniform nanowires under the irradiation of subsequent laser pulses. By controlling the polarization during processing, free direct writing and processing of arbitrary planar patterns can be realized. The length, pitch, and trajectory of the nanogrooves are shown in Figure 6C. Due to the strong near-field localization of the pulse energy within the nanowire, the crosstalk between adjacent grooves can be effectively eliminated. From this principle, the nanowire shown in Figure 6C can be written, which makes it possible to modify birefringence with programmable azimuth and retardance at the nanoscale. Figure 6D shows the uniform retardance of the free-form mode, as well as the orientation of the slow axis and SEM images.

Femtosecond laser induced nanograting structures on the surface and inside of materials have attracted much attention in the field of femtosecond laser-matter interaction research. It is reported that material properties and their dependence on laser excitation determine the morphology of nanogratings. Sapphire is a widely used material in extreme conditions, such as aerospace and chemical industries. As shown in Figure 6E, the process of inducing self-organized nanogrid structures using a femtosecond laser is demonstrated inside a sapphire crystal with some similarities to quartz glass, also as a result of multi-pulse accumulation [47]. We reveal the formation process of a nanogrid inside the sapphire, where initially one-hundred pulses appear with discrete nanoscale holes and defects near the laser focus, and as the pulses increase to 800, these nanoscale holes undergo self-organized joint growth, showing certain periodic nanoscale, and finally, when the number of laser pulses increases to 2000, a regular short-period nanogrid appears in the whole laser processing region. The formation process of nano-grating starts with the appearance of voids in the laser irradiated sapphire, while nano-slits appear at the top and bottom of the voids. Then, these nano-slits act as seeds for the nano-grating. It is important to emphasize one point: the voids of the figure are amorphous materials plus nanomicrospheres in the center of the focus; however, there is still confusion about their physical origin, but we infer that equipartition excitations are excited and produce nanogranules under the local effect of stress-induced nanoslits [26,50,51,52,53,54]. Once nanogridges are formed, they will extend to all laser-modified regions.

The birefringence intensity also varies significantly at different stages of nanogrid formation. The induced birefringence intensity increases gradually with the number of pulses before the nanoscale pores will undergo a self-organized joint growth phase. After the appearance of regular nano-gratings, the birefringence intensity slowly increases to a certain value and then tends to saturate. Since sapphire is a crystalline material, the range of formation parameters of nanograting is much narrower compared to that of glass materials, and the formation region of nanograting is also accompanied by the appearance of larger stresses, and the material is prone to losses, such as cracks, and the birefringence properties generated by nanograting are much smaller than those of the stresses that accompany it.

In order to achieve the target retardation of 633 nm/4 = 158 nm, a five-layer structure was fabricated. During this process, a huge stress-induced birefringence appeared on the boundary, the retardance of which of was 1.5 times that of the scanned area in a single layer (exceeded 266 nm in five layers). To solve the problem of excessive edge stress, the thermal annealing has been adopted to reduce the stress at the edge of the device, and it greatly improves the phase distribution (Figure 6F).

## 7. Conclusions

In conclusion, this review illustrates a broad spectrum of materials that researchers applied in femtosecond laser direct-writing fabrication, for instance, femtosecond laser direct-writing photoresists, femtosecond laser ablation of thin metal films, and femtosecond laser-induced nanogratings. The polarization converter and geometric phase devices, including low-loss Type X structure, were fabricated based on the properties of different materials thanks to the high-precision characteristics of femtosecond laser direct-writing technology. Achromatic feature of polarization converter, speed, and precision of glass internal processing, and the problem of debris are also key issues to improve the fabrication efficiency and performance of devices. When fabricating devices using femtosecond laser-induced nanogratings, the researchers also proposed various formation theories through experimental observations. However, there is still a lack of enough evidence to explain the mechanism, which needs to be further explored to boost the future femtosecond lasers fabrication technique.

## Figures and Tables

**Figure 1 nanomaterials-13-01623-f001:**
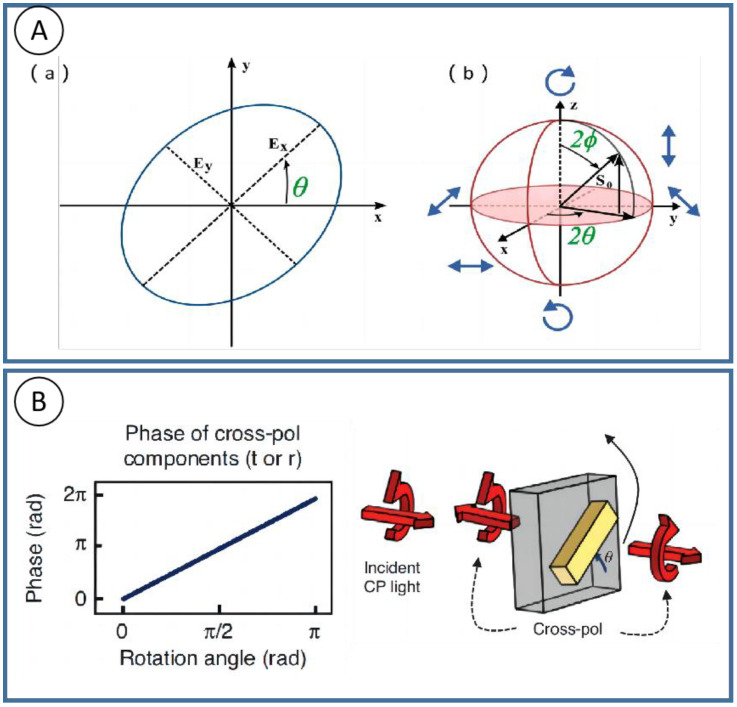
Phase modulation methods and schematic diagram of geometric phase element. (**A**) (**a**) Polarization ellipse and (**b**) Poincare sphere. (**B**) Geometric phase elements empowered by gratings [15].

**Figure 2 nanomaterials-13-01623-f002:**
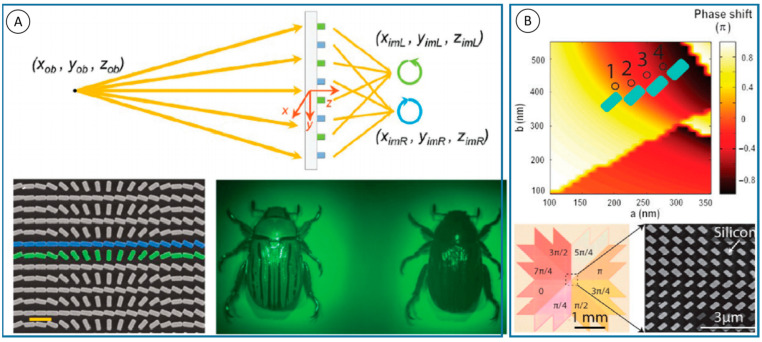
Examples of geometric phase elements. (**A**) Top: schematic diagram illustrating the imaging principle of the PB lens where left circular polarization (LCP) and right circular polarization (RCP) light from the same object at the same coordinates are focused into two spots, respectively. Bottom left: scanning electron microscopy image of the fabricated lens. Scale bar: 600 nm. Bottom right: experimental results [16]. (**B**) Dielectric meta-reflect array for broadband linear polarization converter and optical vortex generation [18].

**Figure 3 nanomaterials-13-01623-f003:**
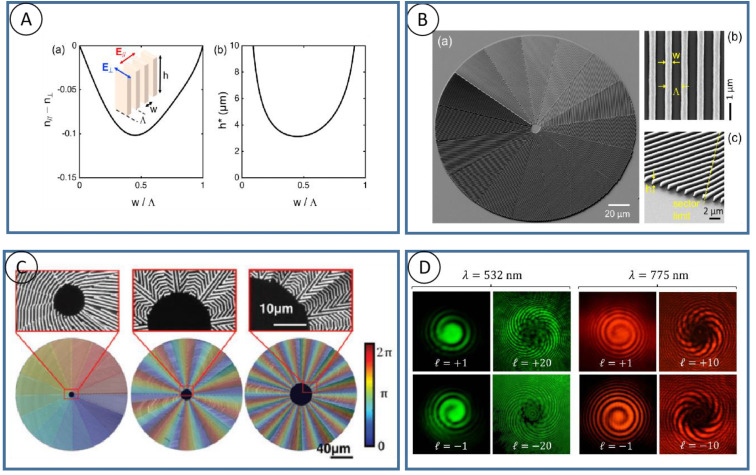
Femtosecond laser direct wiring of geometric phase elements. (**A**) (**a**,**b**) The birefringence of grating structure and the optimized height for Δ = π [28]. (**B**) (**a**) The scanned electron microscope images, (**b**,**c**) magnification of detail [28]. (**C**) Geometric phase plates generate high orders of optical vortex beams fabricated by femtosecond laser writing [28]. (**D**) The Single-beam interference patterns resulting from the q-plates [28].

**Figure 4 nanomaterials-13-01623-f004:**
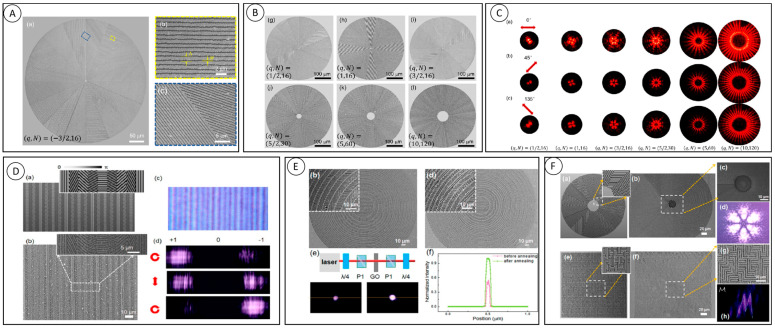
Femtosecond laser ablation on metal film. (**A**) The SEM image of femtosecond laser ablation pattern [36]. (**B**) SEM images of ablated q-plates with different q values. (**C**) Intensity readout of different q-plates with linearly polarized beam [36]. (**D**) Characterization of geometric phase blazed grating [37]. (**E**) Schematic diagram, SEM image and characterization of the dielectric gradient metasurface lens [37]. (**F**) Design and characterization of q-plates and Fourier holograms [37].

**Figure 5 nanomaterials-13-01623-f005:**
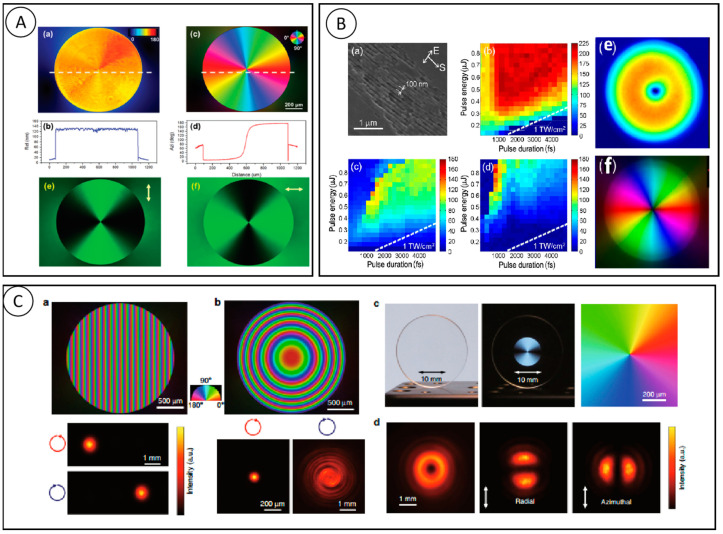
Femtosecond laser-induced nanogratings for polarization convertors and geometric phase elements. (**A**) Radial and azimuthal polarization optical vortex converter. (**a**) Imaged retardance distribution and (**b**) azimuth of slow axis orientation specifies imprinted element as a quarter wave plate; (**c**) Retardance and (**d**) azimuth profiles; (**e**,**f**) Images of polarization sensitive element [39]. (**B**) (**a**–**d**) SEM and birefringence measurements of AF32 glass; (**e**,**f**) quantitative birefringence and diameter polarization converter [45]. (**C**) (**a**,**b**) Birefringence image and light intensity patterns of GP prism and GP lens; (**c**,**d**) vector beam converter without and with a polarizer under linearly polarized white light illumination and the slow axis distribution, as well as intensity pattern [23].

**Figure 6 nanomaterials-13-01623-f006:**
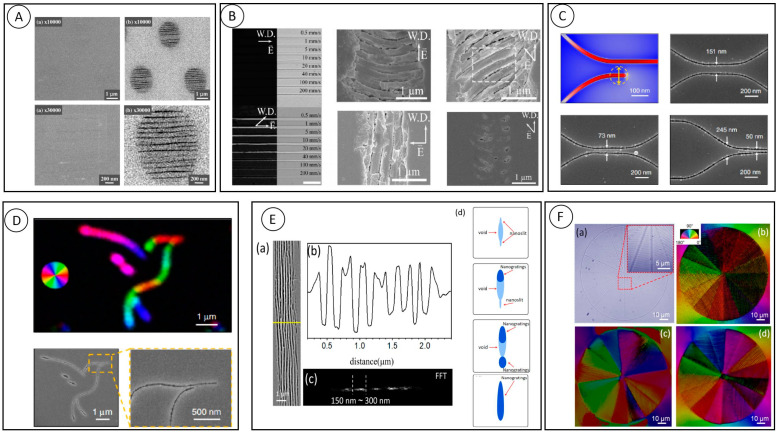
Femtosecond laser-induced nearfield enhancement for the nanograting formation. (**A**) The first nanograting structure found inside glass [32]. (**B**) Optical and cross-polarizer images and SEM images [46]. (**C**) SEM images of machining curvature and separation control [26]. (**D**) Slow-axis orientation maps (pseudo color, Abrio) and SEM images of the free-form written nanogrooves [26]. (**E**) (**a**,**b**) Side-viewed SEM images of laser-induced structures and period estimation; (**c**) FFT analysis diagram; (**d**) Nanogratings periodic variation pattern. (**F**) Characterization of q-plate before and after annealing, (**a**) q-plate optical microscope photograph. (**b**) single-layer q-plate birefringent microscope photograph. (**c**) Color slow axis orientation distribution of 5-layer q-plate nanogratings. (**d**) Color slow-axis orientation distribution of 5-layer q-plate nanogratings after constant thermal annealing.

## Data Availability

Not applicable.

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
