# Peer review of "Recent Developments of Femtosecond Laser Direct Writing for Meta-Optics"

_nanomaterials, 2023, doi:10.3390/nano13101623_

Round 1
Reviewer 1 Report
The manuscript “Recent developments of femtosecond laser direct writing for meta-optics” is well written.
However, I have two remarks:
1. As it is a review paper, so the authors must insert the Reference every time they discuss the certain problem, on which other research teams have worked, like in:
- 3. Fabrication of spin-orbit angular momentum couplers by femtosecond laser two-photon direct writing technique
- 4. Femtosecond laser ablation of gold films
- 5. Femtosecond laser direct writing of glass-induced nanogratings
- 6. Nanograting induction mechanism
2. The authors should take in account more European and American research teams working on the corresponding fields and to include them in the References list, since in this variant the Reference list seems uncompleted.
Author Response
Response to 1 : Thanks. We have revised the article following the comments.
Response to 2: Thanks. We have added a part of the content in the text , while also adding some citations to the references.

Reviewer 2 Report
Dear Authors,
You provided a balanced overview of the subject area. The manuscript's material is presented transparently and quite logically. In conclusion, I would like to see your opinion, which will help develop the direction both on a practical level and fundamental one.
Good luck
Author Response
Thanks for the overall comments.

Reviewer 3 Report
The review presents technologies of exposure to laser femtosecond radiation on various materials for laser direct writing fabrication, ablation of thin metal films and laser induced nanogratings. The review is well written and illustrated. However, its content needs to be improved. The main question is for what reader is this review prepared? If for those who are professionally engaged in laser femtosecond modification of the properties of materials, then this review does not contain original materials. The authors included in the review the results of their experiments, but they have already been previously published in [Xu, S.; Fan, H.; Xu, S.-J.; Li, Z.-Z.; Lei, Y.; Wang, L.; Song, J.-F. High-efficiency fabrication of geometric phase elements by femtosecond-laser direct writing. Nanomaterials 10 (2020) 1737; Xu S. et al. Ultrafast laser-inscribed nanogratings in sapphire for geometric phase elements. Optics Letters 46 (2021) 536]. In the event that the review is intended for those who only intend to use this technology, then this review needs to be finalized.
I believe that in the introduction it is necessary to discuss three main regimes of laser material modifications, which are accessed depending on whether the pulse duration is shorter, comparable or longer than the time required by the hot electrons to transfer their energy to the lattice via electron-phonon scattering, i.e. femto-, pico and nanosecond laser pulses.
I am agreeing that femtosecond laser processing for a number of applications and materials has its undeniable advantages. However, it is necessary to discuss the limitations of laser excitation by femtosecond pulses, which lead to strong non-equilibrium conditions as electrons in the conduction band are heated much faster than they can cool by phonon scattering. The absence in the review of a small section on the limitations of femtosecond laser processing may mislead the readers of this article. In particular, you can discuss the fundamental limitations of in-volume laser direct writing of crystalline silicon [Chambonneau M. et al. Laser & Photonics Reviews 15 (2021) 2100140], problems in manufacturing of microlens arrays on soda-lime glass substrates by using a femtosecond laser [Delgado T et al. Optics and Lasers in Engineering 86 (2016) 29], or in the producing of uniform quasi-periodic structures on noble metals (e.g. copper) [Anderson M. et al. Surface and Coatings Technology 409 (2021) 126872] and other cases.
On the other hand, the review could be supplemented with one or two paragraphs on the advantages of using femtosecond laser radiation for surface and bulk modification of diamond, a superhard and chemically resistant material. A significant part of the above is not so much a negative remark, but rather an advice to the authors of the manuscript.
Author Response
First of all, I would like to thank the reviewer for the article. I have read it carefully. The limitations of femtosecond laser processing are undeniable, but femtosecond lasers have unique advantages over other processing methods (e.g., nanosecond lasers) in the processing of high-precision geometric phase components. In the introduction I have added a section where I emphasize the unique advantages of femtosecond lasers for high precision processing. This will make the reader more aware of where the article is pointing. At the same time, the introduction has added the discussion on the modification mechanism of laser and material.
Next, this paper focuses on the local electromagnetic modification of ultramicroscopic optical devices using micro-nano structures. Based on the preparation of micro and nano structures, this paper reviews the research progress in the preparation of polarization conversion devices and geometric phase devices by femtosecond laser processing techniques. Laser femtosecond modification of material properties is not the subject of this paper. This paper is intended for those who intend to induce nanogrid structures and fabricate geometric phase devices using femtosecond laser technology only. I will put these articles you have provided in my references.
Finally regarding diamond, there have been researchers who induced nanostripes on diamond surfaces, but no one has prepared the corresponding geometric phase devices. The fabrication of geometric phase devices is a huge challenge. Since this article is mainly about geometric phase devices, I do not think it is necessary to discuss diamond in the article now.
